# What is the nature of motor adaptation to dynamic perturbations?

**Etienne Moullet**[ID]**, Agnès Roby-Brami, Emmanuel Guigon**[ID]*

Sorbonne Université, CNRS, Institut des Systèmes Intelligents et de Robotique, ISIR, Paris, France

* emmanuel.guigon@sorbonne-universite.fr

## Abstract

When human participants repeatedly encounter a velocity-dependent force field that distorts their movement trajectories, they adapt their motor behavior to recover straight trajectories. Computational models suggest that adaptation to a force field occurs at the action selection level through changes in the mapping between goals and actions. The quantitative prediction from these models indicates that early perturbed trajectories before adaptation and late unperturbed trajectories after adaptation should have opposite curvature, i.e. one being a mirror image of the other. We tested these predictions in a human adaptation experiment and we found that the expected mirror organization was either absent or much weaker than predicted by the models. These results are incompatible with adaptation occurring at the action selection level but compatible with adaptation occurring at the goal selection level, as if adaptation corresponds to aiming toward spatially remapped targets.

## Author summary

Motor adaptation is a fundamental component of the acquisition and maintenance of skilled behaviors. Yet the nature of motor adaptation remains poorly understood: when we encounter forces which repeatedly perturb our movements, do we change our actions or our plans? Current computational models of motor control favor the former, but this assumption has not been thoroughly investigated. To address this issue, we compared predictions of a model of motor adaptation based on changes at the action level with observations obtained from a group of human participants involved in a motor adaptation task. The behavior of the participants clearly differed from the model's predictions. These results challenge contemporary perspectives on motor adaptation.

## Introduction

Motor behavior is both highly stable and widely flexible [1–3]. On the one hand, a large repertoire of skilled, efficient behaviors (e.g. speech production, handwriting, gait, . . .) is maintained for decades, often robust in the face of injury, aging, disease or brain damage. On the other hand, a few movements performed in a novel sensorimotor environment (e.g. wearing prismatic glasses, holding a visco-elastic manipulandum, . . .) or in some altered physiological

**Data Availability Statement:** Data files are available from osf (459.6 Mo): https://osf.io/bwcp9/?view_only=7180abd47a44479cbfe4e4578161f627.

**Funding:** The author(s) received no specific funding for this work.

**Competing interests:** The authors have declared that no competing interests exist.

state (e.g. muscular fatigue, pain, . . .) can induce lasting changes in motor performance [4–7]. A proper balance between stability and flexibility is necessary so that (1) ingrained skills remain sensitive to steady and persistent changes in the environment, the body and the nervous system but are not disproportionately influenced by temporary, incidental events; and (2) new skills can develop at any time. How then is skilled movement organized in response to these contrasting priorities?

Motor learning and skill acquisition are generally understood from two distinct viewpoints [3]. The first view holds that learning occurs at the action selection (control) level, and modifies the mapping between the intended goals and those actions inclined to achieve these goals (Fig 1, *purple*). For instance, in the typical laboratory example of adaptation to a velocity-dependent force field (dynamic perturbation; [4]), learning has been described either as a compensation process, i.e. mapping is learned between states and compensatory forces opposite to the applied forces ([4]; Fig 1B, *left* and *center*; see also Fig 1C, *left* and *center* for the case of a visuomotor rotation), or as a reoptimization process, i.e. mapping is learned between goals and optimal forces to achieve the goals in the presence of the applied forces [8]. According to the second view, learning occurs at the goal selection level and modifies the mapping between intended and actual goals irrespective of how to achieve these goals (Fig 1B, *right*). For instance, adaptation to a visuomotor rotation of the visual display (kinematic perturbation) results from a redirection process, i.e. a remapping between target and movement vectors ([9]; Fig 1C, *right*). Although the latter learning process appears more flexible and frugal than the former, it is unclear whether it can account for adaptation to dynamic perturbations, i.e. when new patterns of force need to be learned.

Models based on compensation or reoptimization are well formulated models that can be used to make predictions on adaptation to dynamic perturbations (velocity-dependent force fields; [4,8]). In particular, the shape of after-effect trajectories, i.e. late trajectories in the absence of the force field after adaptation, should incorporate a "negative image" of the forces induced by the applied force field, a reflection which mirrors before-effect trajectories, i.e. early trajectories in the presence of the force field before adaptation (this is exactly the case for the compensation model; [4]). The shape of before-effect trajectories has been thoroughly documented. They are initially curved "away" from the baseline (unperturbed) trajectory with a late ensuing correction toward the target [4]. We have not identified any study that quantitatively documents the shape of after-effect trajectories. Yet qualitative observations on published figures suggest that after-effect trajectories do not obey the predicted mirror organization (Fig 2 in [10]; Fig 4 in [11]; Fig 1 in [12]; Fig 2 in [13]; Fig 2 in [14]; Fig 1B in [15]). In fact after-effect trajectories seem to resemble "kinematic" trajectories, i.e. trajectories observed during visuomotor rotation or target jump tasks rather than "dynamic" trajectories observed during force field tasks (examples of contrast between kinematic and dynamic trajectories in Fig 2 in [16]; Fig 6 in [17]). They might thus be compatible with a redirection process, as if adaptation corresponded to aiming toward spatially remapped targets.

The main goal of this study is to clarify the nature of before-effect and after-effect trajectories during a force field adaptation task in order to assess the pertinence of the compensation/reoptimization process as a basis for motor adaptation. A secondary goal is to promote the redirection process as a promising candidate for motor adaptation.

## Results

We designed a force field adaptation experiment with a large number of trials and a small fraction of catch trials (unexpected addition or removal of the force field) to obtain "pure" before-effect and after-effect trajectories uncontaminated by ongoing learning processes [18]. Twenty

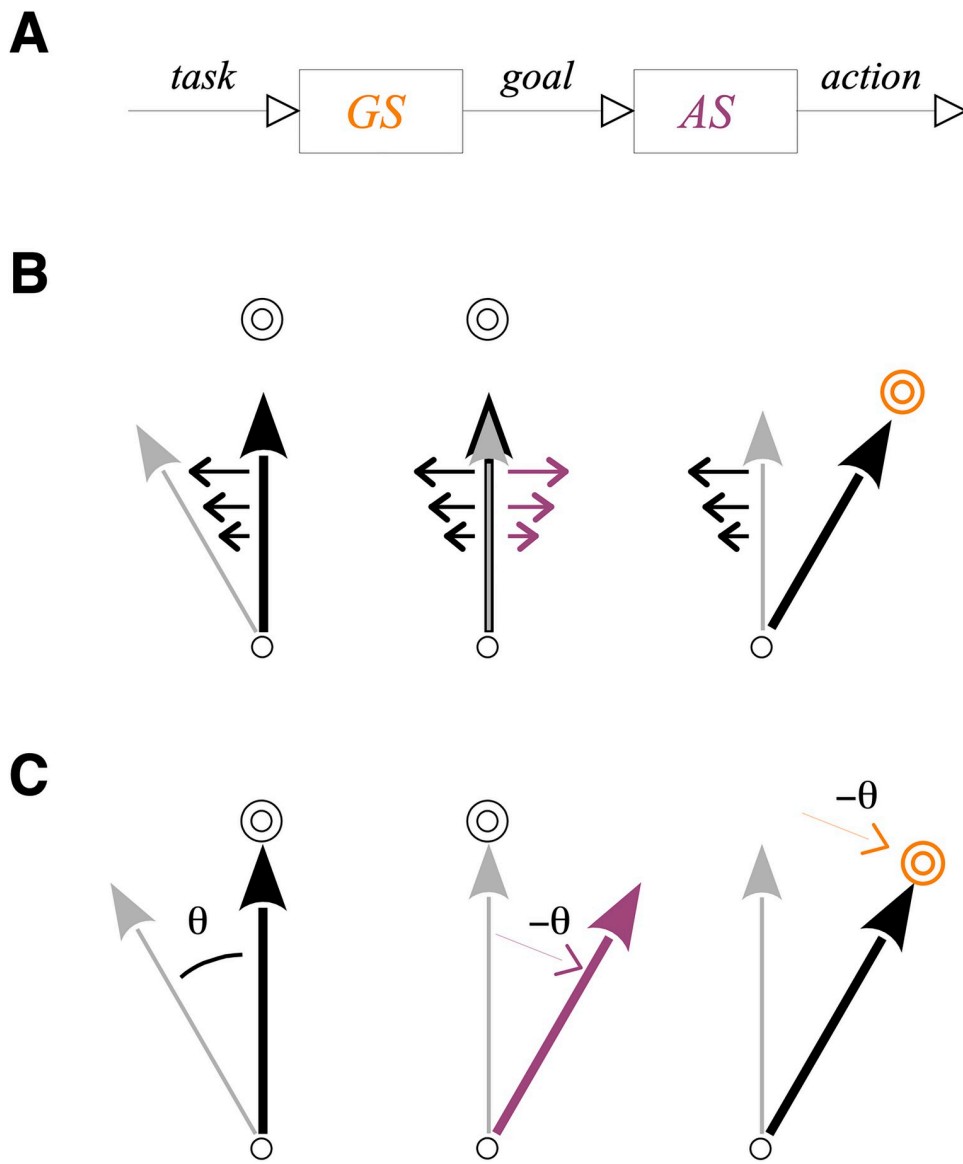

**Fig 1. Goal selection vs action selection. A.** The motor system contains: (1) a process, called action selection (*AS*; *purple*), which translates a current goal (e.g. a target to reach) into the proper displacement of the current effector (e.g. the arm) toward the goal (the target); (2) a process, called goal selection (*GS*; *orange*), which provides the current goal for a given task. **B.** Schematic of adaptation to a force field perturbation (only the early phase of movement is described). The small circle is the starting position, the double circle the goal position, the black arrow the planned displacement, and the gray arrow the actual (or observed) displacement. (*left*) For a planned displacement toward the goal position, the force field (*black* leftward arrows) induces an initial actual displacement in the direction of the perturbation. (*center*) Adaptation at the *AS* level consists in keeping the same goal position and applying compensatory forces (*purple* rightward arrows). (*right*) Adaptation at the *GS* level consists in re-aiming toward a new goal position (*orange* double circle). **C.** Schematic of adaptation to a rotation of the visual display. (*left*) For a planned displacement toward the goal position, the rotation induces an initial actual displacement in the direction of the rotation. (*center*) Adaptation at the *AS* level consists in keeping the same goal position and applying compensatory rotation (purple rightward arrow). (*right*) Adaptation at the *GS* level consists in re-aiming toward a new goal position (*orange* double circle).

two participants were asked to make fast, planar, forward arm reaching movements from a start position to a target position located 0.1-m away in the presence of a null field or a perpendicular clockwise (CW) or counterclockwise (CCW) velocity-dependent force field (Fig 2A

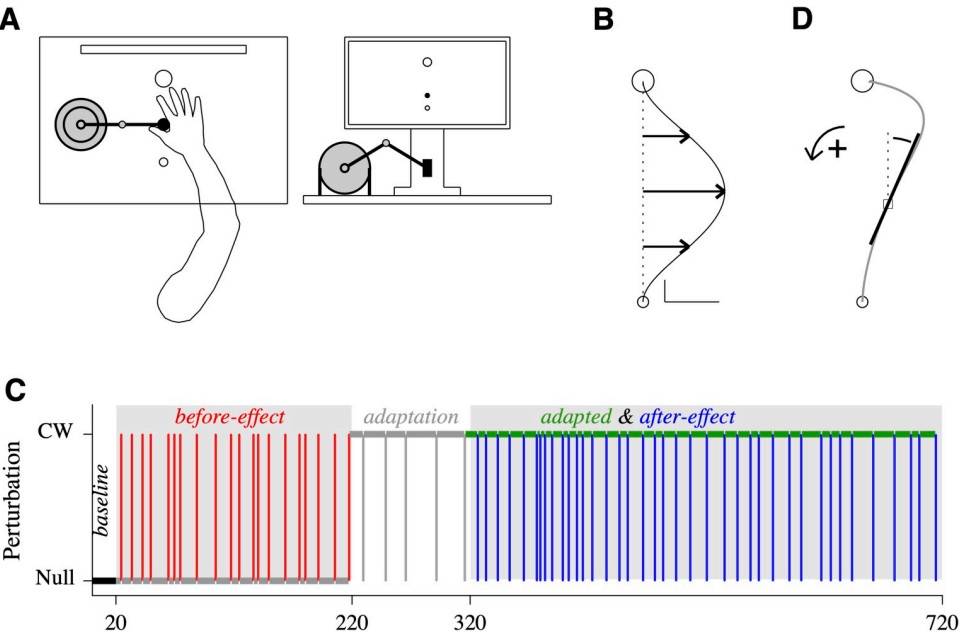

**Fig 2. Description of the experiment. A**. Experimental setup. (*left*) Top view. The *small open circle* is the start position and the *large open circle* the target position. The *black circle* is the robot handle. The *elongated open rectangle* is a top view of a monitor. (*right*) Front view. The start position, target position, and visual feedback of hand position (*black circle*) are shown on the monitor. The *black rectangle* is the robot handle. The scales are not respected. **B**. Simulated velocity-dependent force field. A minimum-jerk velocity profile with a 0.3 m/s peak was multiplied by a 5 N/m force field. Vertical scale: 0.01 m. Horizontal scale: 1 N. **C**. Experimental protocol. The force field level (null or CW) is indicated by the horizontal *black* (baseline block), *gray* (before-effect and adaptation blocks) or *green* (adapted and after-effect blocks) *thick* line segments. The vertical line segments indicate catch trials: unexpected CW force field in the before-effect block (*red*); unexpected null force field in the adaptation block (*gray*) and in the after-effect block (*blue*). Only the colored trials (*black*: baseline; *red*: before-effect; *green*: adapted; *blue*: after-effect) were analyzed. **D**. Graphical definition of the trajectory angle. At one point along the trajectory (*open square*), the trajectory angle is the angle between start position/target position direction (*dashed line*) and the tangent to the trajectory (*thick line*).

and 2B). The participants performed four blocks of trials (Fig 2C) and we identified baseline, before-effect, adapted, and after-effect trajectories (see **Material and methods**). For data analysis, all trajectories were displayed with a CW deviation, i.e. for a CCW perturbation, a vertical symmetry was applied to the trajectories. A trajectory was described by (1) the angle (counted positive in the CCW direction) of its tangent relative to the target direction (Fig 2D); (2) the time derivative of the trajectory angle (see **Material and methods** for details).

The same color code is used in all the figures: *black* (baseline), *red* (before-effect), *green* (adapted), *blue* (after-effect). For legibility, a * is added on each figure panel when the display contains results of simulations.

## Predictions

The compensation model [4] makes immediate predictions on the shape of before-effect and after-effect trajectories and corresponding velocity profiles (S1 Fig). These prediction will not be further considered: they are robust but lack pertinence as the compensation model is not a general model of motor control (see **Discussion**). In order to build precise predictions for the reoptimization model, we proceeded in the following way. We considered one participant (P7) and analyzed detailed characteristics of her motor behavior (Fig 3). We calculated the mean baseline and before-effect trajectories (Fig 3A) and velocity profiles (Fig 3B). For each single trial (e.g. a baseline trial; Fig 3C), we calculated a discrete measure of the frequency content

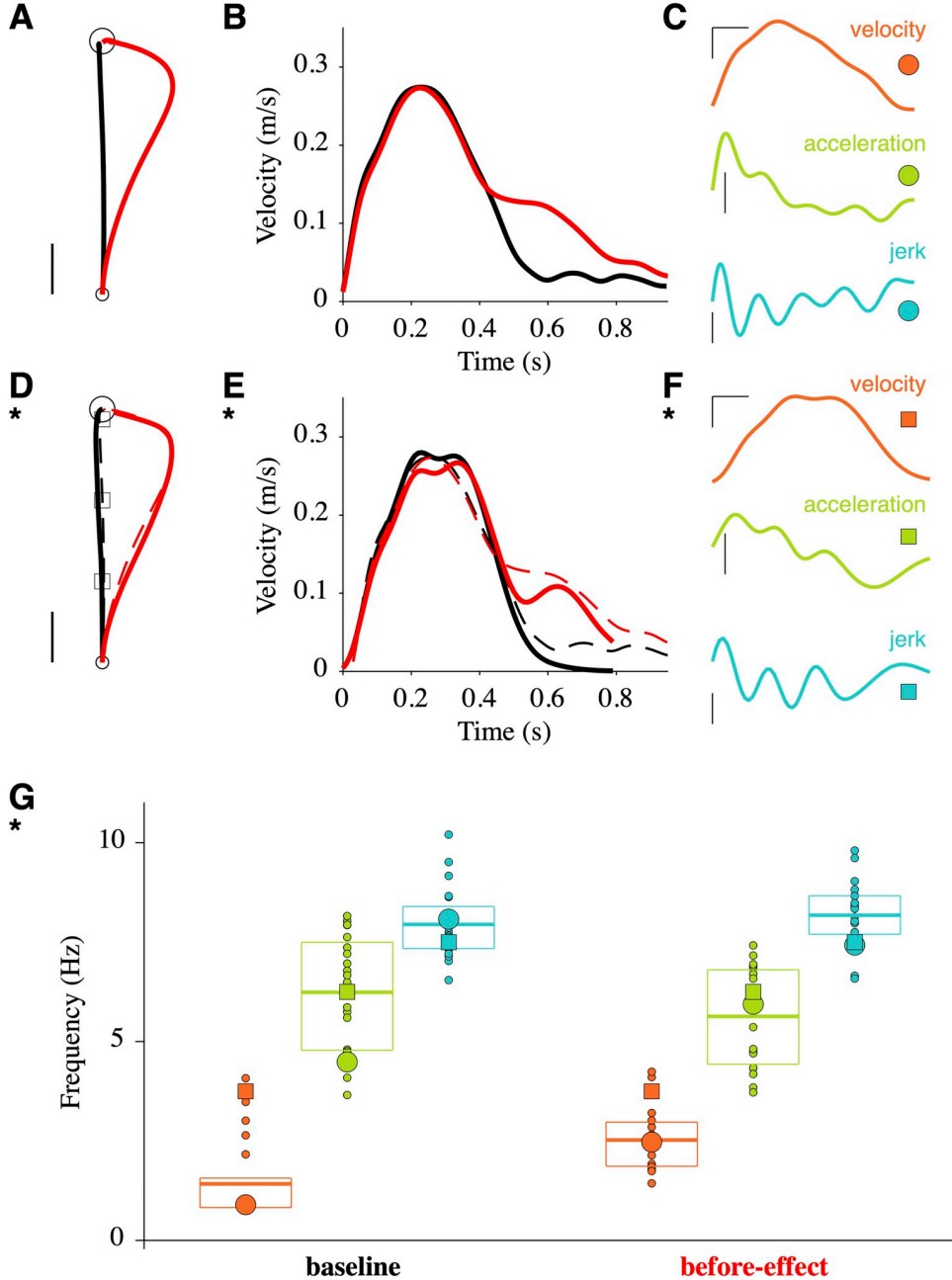

**Fig 3. Model adjustment based on data of participant P7. A**. Mean baseline (*black*; 17 trials) and before-effect (*red*; 19 trials) trajectories for P7. Scale: 0.02 m. **B**. Mean velocity profiles of baseline and before-effect trajectories for P7. **C**. Velocity (scale 0.1 m/s), acceleration (scale 2 m/s$^2$) and jerk (scale 30 m/s$^3$) profiles for a single baseline trial (P7). Time scale: 0.1 s. **D**. Simulated baseline (*plain black*) and before-effect (*plain red*) trajectories compared to experimental trajectories (*dashed*; data from **A**). Squares are via-points for the simulated trajectories. Same scale as in **A**. **E**. Simulated velocity profiles. **F**. Velocity (same as *black* in **E**), acceleration and jerk profiles for the simulated baseline trajectory. The profiles have been truncated to match the duration of the trial in **C**. Same scales as in **C**. **G**. Peak frequency for velocity (*orange*), acceleration (*light green*) and jerk (*light blue*) profiles for individual trials (*small dots*). *Large circles* correspond to the trial in **C**. *Large squares* correspond to the simulated trial in **F**. *Thick lines* are mean values and *boxes* indicate 25–75 percentiles.

(peak frequency; number of minima+number of maxima/duration/2) of velocity, acceleration and jerk traces. We plotted the peak frequency of all trials for the two types of trial (Fig 3G). These results show that the smoothness of mean trajectories and velocity profiles (Fig 3A and 3B) is an artifact of averaging widely nonsmooth and variable single trials (Fig 3C and 3G). Although these observations are not surprising [19,20], they cannot be explained by models that produce temporally invariant smooth movements [21–23]. To circumvent this difficulty, we considered a model which explains the frequency content of movements (Fig 3C and 3G) by the pursuit of intermediate goals (via-points) updated at ~8 Hz (see **Material and methods**; [20,24]. We searched for a series of via-points $S$ and model parameters that account for experimental paths and velocity profiles of baseline and before-effect trajectories (Fig 3D and 3E; for a parametric study of the model, see below). The series $S$ contained three intermediate via-points (*squares*; Fig 3D) at 32, 64 and 96% of the distance to the target in the direction of the target, and the target itself (*circle*; Fig 3D). Note that we did not search for the "best fit", as all single trials were different (Fig 3C and 3G). Note also that the intensity of the modeled force field ($\varphi$) was lower than that of the experimental field (see **Discussion**). The amplitude and frequency contents of the resulting movement were consistent with the experimental data (Fig 3F and 3G). At this stage, the proposed model is appropriate for trajectory formation and online motor control during perturbations and further accounts for many characteristics of motor behavior [24]. We can now obtain proper predictions for the reoptimization model (Fig 4). The adapted trajectory was not a straight path but an overcompensation (*green*; Fig 4A) which is consistent with [8]. Its velocity profile was close to the baseline velocity (*green* vs *black*; Fig 4B). The after-effect trajectory had the expected mirror organization relative to the before-effect trajectory (*blue* vs *red*; Fig 4A) and a velocity profile which resembled the before-effect profile (*blue* vs *red*; Fig 4B). The mirror effect is quantitatively described in Fig 4C and 4D. The trajectory angles had opposite monotonic trends for before-effect and after-effect trajectories over the first ~0.6 s (*blue* vs *red*; Fig 4C) with corresponding changes in the sign of the derivatives (*blue* vs *red*; Fig 4D). In the following, we will focus on the early part of the trajectories (0.4 s; *dotted boxes* in Fig 4C and 4D; 4E and 4F) since trajectory averaging for experimental data may produce unreliable results for the late part of the trajectory. Two quantitative observations are relevant: (1) the angle derivative of the before-effect trajectory became positive at 0.29 s (*vertical red dashed line*; Fig 4F). This result is consistent with experimental data in P7 and across all the participants (S2 Fig); (2) the angle derivative of the after-effect trajectory became negative at 0.31 s (*vertical blue dashed line*; Fig 4F) which means that the derivative is negative 22.5% of the time during the first 0.4 s. For comparison with experimental data, we will use this number rather than the time of change in sign which might not be well defined in the data (e.g. due to multiple changes in sign).

On the one hand, the expected positive sign of the derivative of the after-effect trajectory angle would add support to the reoptimization model. On the other hand, a null or negative derivative would contradict the reoptimization model.

## Two participants

Results for participant P7 shown in Fig 5 (same format as in Fig 4) followed the typical pattern observed in force-field adaptation experiments [4,8]: 1. The mean baseline trajectory was straight (*black*; Fig 5A); 2. The mean before-effect trajectory deviated in the direction of the perturbation with a late hook-like correction (*red*; Fig 5A); 3. The mean adapted trajectory was straighter than the mean before-effect trajectory but not as straight as the baseline trajectory (*green*; Fig 5A); 4. The mean after-effect trajectory was deviated in the direction opposite to the perturbation (*blue*; Fig 5A); 5. The velocity profiles had a large initial peak followed by one or

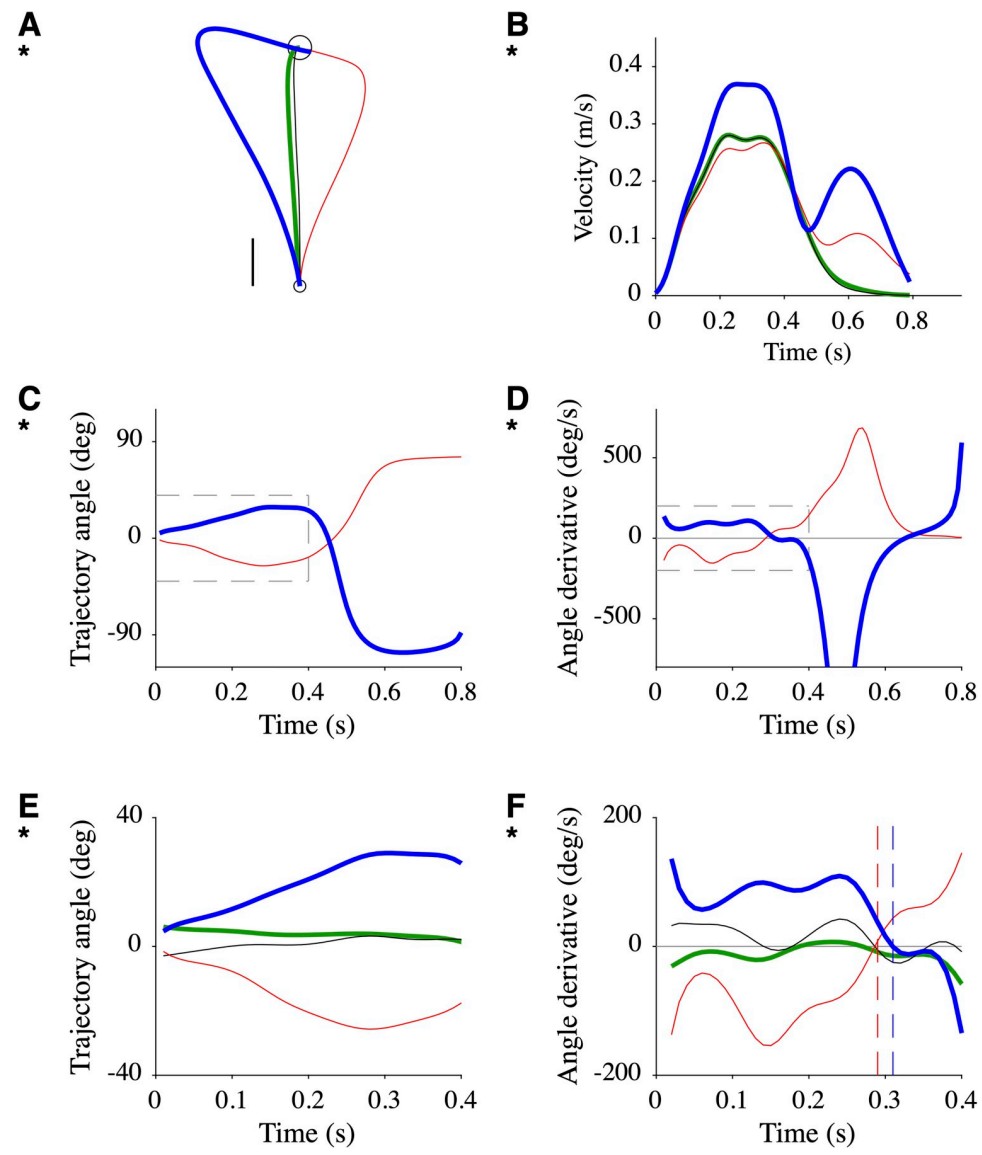

**Fig 4. Model predictions. A**. Simulated adapted (*green*) and after-effect (*blue*) trajectories corresponding to simulated baseline (*black*) and before-effect (*red*) trajectories shown in Fig 3D and reproduced here with *thin lines*. Scale: 0.02 m. **B**. Simulated velocity profiles. **C**. Trajectory angle for **A**. **D**. Trajectory angle derivative for **A**. **E**. Zoom on trajectory angle (*dotted box* in **C**). **F**. Zoom on trajectory angle derivative (*dotted box* in **D**). *Vertical dashed lines* indicate the time of change in derivative sign.

more smaller peaks (Fig 5B); 6. Single trial trajectories were variable but consistent with the mean trajectory (Fig 5C); 7. Lateral deviation decayed exponentially across trials ($R^2 = 0.69$; Fig 5D).

To test the reoptimization model, we analyzed the time course of the mean trajectory angle (Fig 5E) and trajectory angle derivative (Fig 5F). As expected, the mean angle of the before-effect trajectory decreased until ~0.3 s and then increased (*red*; Fig 5E and 5F and inset). The mean angle of the after-effect trajectory was initially approximately constant and then decreased (*blue*; Fig 5E and 5F and inset). To assess the statistical significance of this observation, we performed a *t*-test on the sign of the angle derivative ($H_0$: = 0 vs $H_1$: ≠ 0; $N = 34$ trials) at each timestep. The corresponding *p*-value was >0.05 for the first 0.1 s (Fig 5G), which

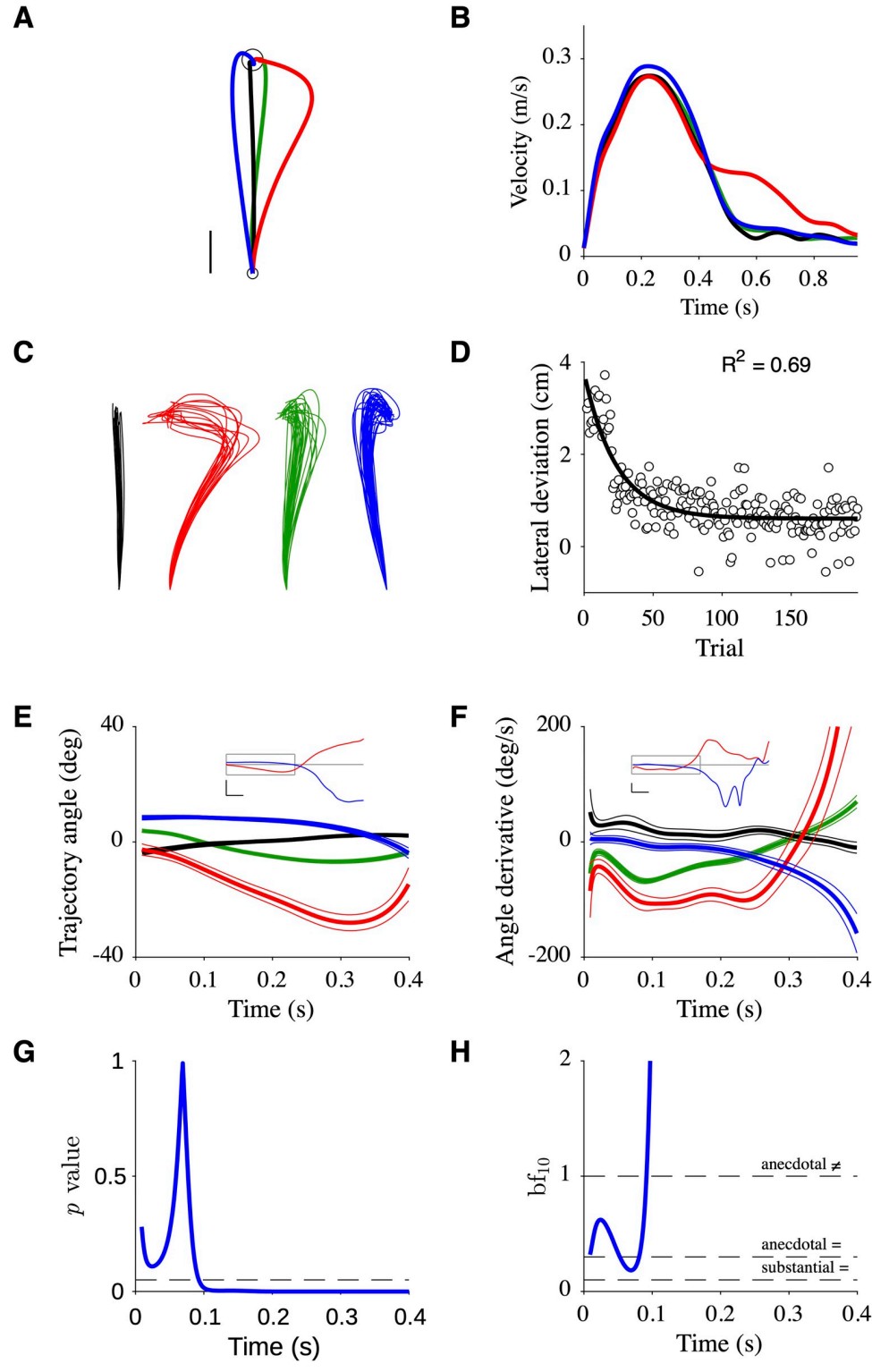

**Fig 5. Data of participant P7. A**. Mean trajectories. Scale: 0.02 m. **B**. Mean velocity profiles. **C**. Single trials (17, 19, 107, 34 trials, from left to right). Same scale as in **A**. **D**. Changes in lateral deviation (maximum of trajectory deviation from the start position/target position line) with training. The data were taken from **C** (*red* and *green*). Fitting an exponential decay is shown. **E**. Mean trajectory angle over the first 0.4 s with the 95% confidence interval. Inset: mean trajectory angle over the first 0.8 s; the box indicates the 0–0.4 s window. Scale: 0.1 s, 60 deg. **F**. Same as **E** for mean trajectory angle derivative. Scale: 0.1 s, 200 deg/s. **G**. *p*-value of a test ≠0 vs = 0 for trajectory angle derivative in **F**. The

*dotted line* indicates 0.05. **H**. Bayes factor for the test $\neq 0$ vs = 0. The *dotted lines* delimit regions of interpretation of Bayes factors.

indicates that we cannot reject the hypothesis that the angle derivative is zero. The *p*-value was <0.05 after 0.1 s (Fig 5G), meaning that the angle derivative was significantly different from zero and negative. We calculated the Bayes factor $bf_{10}$ for $H_1$ vs $H_0$ which indicated that the data were 1 to 5 times more likely under $H_0$ than under $H_1$ when $p > 0.05$ (Fig 5H).

A different behavior was observed for participant P5 (Fig 6). The mean before-effect and after-effect trajectories were symmetrically organized (Fig 6A and 6C and 6D). A statistical analysis indicated that the angle derivative of the after-effect trajectory was non-zero and positive between ~0.1 and ~0.3 s following movement onset (*p*-value <0.05, Fig 6E; $bf_{10} > 3$, Fig 6G). Although the behavior of this participant matches some predictions of the reoptimization

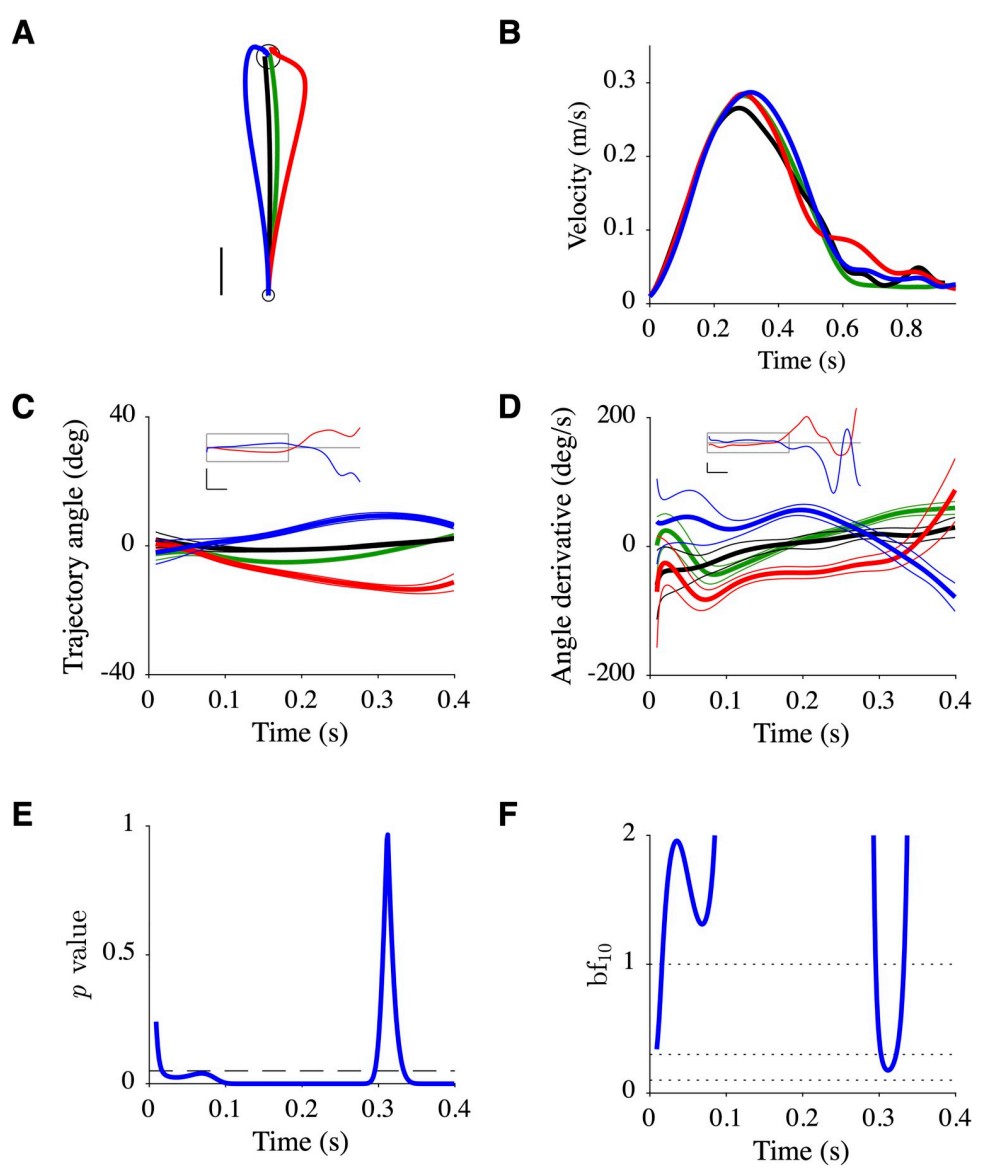

**Fig 6. Data of participant P5.** Same organization as Fig 5 with **C, D, E, F** corresponding to **E, F, G, H**.

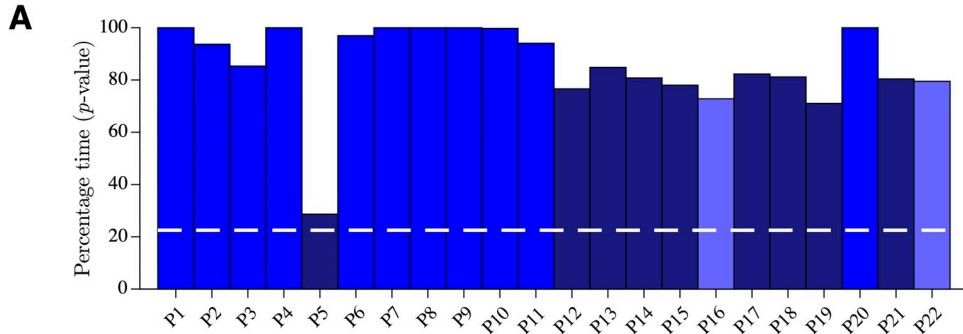

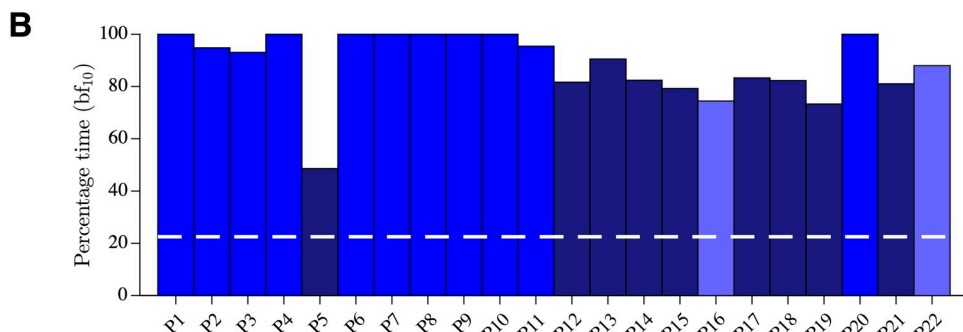

**Fig 7. All participants. A**. Percentage of time over the first 0.4 s during which the angle derivative of the after-effect trajectory was statistically null or negative using *t*-test *p*-values. *Dashed white* line represents model prediction. Color code for the participants: *blue*, behavior incompatible with the reoptimization model; *dark blue*, behavior partially compatible with the reoptimization model; *light blue*, behavior with no effect of training. **B**. Same as **A** using Bayes factor.

model, the experimental path and velocity profile of the after-effect trajectory were different from the predicted path and velocity profile (Figs 6A and 6B vs 4A and 4B).

## All participants

For each participant, we calculated the percentage of time over the first 0.4 s during which the angle derivative of the after-effect trajectory was statistically null or negative using both *p*-values and Bayes factors (see Fig 5G and 5H). The reoptimization model predicts that this percentage should be around 22.5 (Fig 4F). The experimental percentage was different from the predicted percentage for all the participants (Fig 7A and 7B). The behavior of ten participants (*blue bars*; Fig 7) was similar to the behavior of P7 (see Figs 5 and S3). The behavior of eight participants (*dark blue bars*; Fig 7) was similar to the behavior of P5 (see Fig 6). The quantitative results for these participants are shown exhaustively in S4 Fig. It can be observed that the behavior of these participants is rather homogeneous and differs qualitatively and quantitatively from the predicted behavior in terms of path and velocity profile. The two remaining participants (*light blue bars*; Fig 7) failed to improve their behavior with training (S5 Fig).

## Redirection model

We simulated adaptation through redirection using an ad-hoc series of via-points $S'$ to obtain an adapted trajectory (*green*; Fig 8A, *left*) which resembles a real adapted trajectory (*green*; Fig 5C). We generated small variations around these via-points to obtain an ensemble of adapted

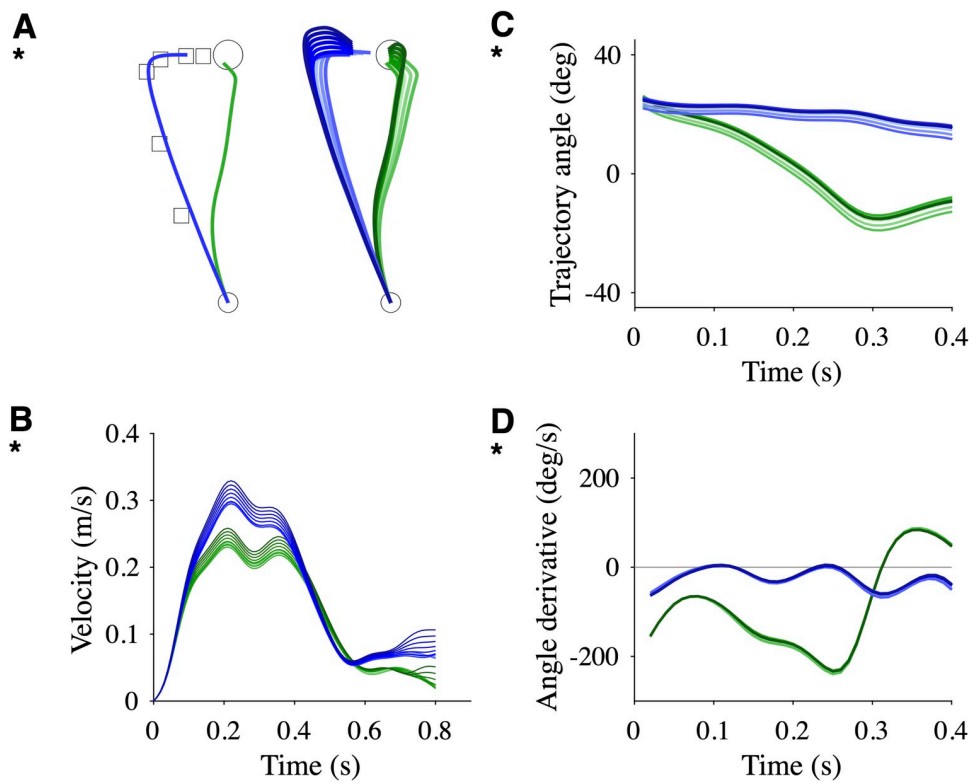

**Fig 8. Simulations of the redirection model. A**. (*left*) Adapted (*green*) and after-effect (*blue*) trajectories corresponding to a series of via-points (*squares*). (*right*) Multiple adapted and after-effect trajectories corresponding to variations of the series of via-points. **B**. Corresponding velocity profiles. **C**. Corresponding trajectory angles. **D**. Corresponding trajectory angle derivatives.

trajectories (*green*; Fig 8A, *right*). The corresponding after-effect trajectories (*blue*; Fig 8A) resembled real after-effect trajectories (*blue*; Fig 5C). The velocity profiles, the trajectory angles and the trajectory angles and the trajectory angle derivatives were consistent (Fig 8B and 8C and 8D). As expected, we observed an absence of mirror effect, the angle derivative of after-effect trajectories being positive < 10% of the time in the first 0.4 s (Fig 8D).

## Parametric study of the reoptimization model

The conclusions of this study are highly dependent on the predictions of the reoptimization model. As the model contains parameters, it is important to understand the influence of these parameters on the proposed predictions. We explored the role of 3 parameters: the feedback delay $\Delta$, the noise ratio $\sigma^{\xi}/\sigma^{\omega}$ of motor to sensory noise variance used in the state estimator, and the muscle gain $g_{sh}/g_{el}$. The first two parameters modulate how sensory information participates in state estimation. The third parameter calibrates the contribution of shoulder and elbow torques to coordination. The trajectories, velocity profiles, trajectory angles and trajectory angle derivatives were consistent across variations of these parameters (S6, S7 and S8 Figs). The mirror organization between before-effect and after-effect trajectories was robustly observed. We note that the time at which the angle derivative of the before-effect trajectory becomes negative (Fig 4F) varied with the feedback delay (S6D Fig) and the torque ratio (S8D Fig).

A set of parameters ($w_\theta$, $w_{\dot{\theta}}$, $w_\alpha$, $w_\varepsilon$) specify the boundary conditions at the via-points, i.e. whether position, velocity, activation and excitation are forced to take specified values. The

predictions were built with $w_\theta \neq 0$ and $w_{\dot\theta} = w_\alpha = w_\varepsilon = 0$ (constraints only on position). The role of these parameters is illustrated in S9 Fig. Although they have little influence on acceleration (S9A Fig), they have a clear effect on jerk, with a higher level of jerk whenever the constraints are not applied exclusively to position (S9B Fig). Only the lower level of jerk (constraints on position) is consistent with experimental data (Fig 4C and 4D).

## Discussion

Classical computational models of motor adaptation assume that learning occurs at the action selection level [4,8]. We derived predictions for these models which show that after-effect trajectories following adaptation to a velocity-dependent force field are close to a mirror of before-effect trajectories (Figs 4 and S1). Experimental data collected in twenty-two participants did not follow these predictions (Figs 5, 6, 7 and S3 and S4). We discuss implications and limitations of these observations.

To open the discussion, we note that we have worked from the modeling prediction that a mirror organization should be observed between before-effect and after-effect trajectories following adaptation to a force field. Yet, at least in the case of adaptation to a visuomotor rotation, the strength and the shape of the after-effects may depend not only on the nature of the perturbation but also on the adaptation protocol, e.g. the presence of error-clamp trials [25]. It is unclear how we can account for this fact in the framework of reoptimization models. We also note that the present study is not concerned with how adaptation occurs on a trial-by-trial basis and with issues related to feedforward and feedback corrections [26].

Hundreds of force field adaptation studies have been performed since the seminal study of Shadmehr and Mussa-Ivaldi (1994) [4], but none of them have quantitatively documented properties of after-effect trajectories. Although many published figures could informally be used to gain qualitative information on before-effect and after-effect trajectories and their differences (to mention a few: Figs 3 and 7 in [27]; Fig 13 in [4]; see **Introduction** for other references), they are not sufficient to draw firm conclusions. The lack of a specific interest for after-effect trajectories might be related to the prevalent view in computational motor control that adaptation results from changes at the control level, and that properties of after-effect trajectories are a direct by-product of these changes [4,8]. In this framework, an after-effect trajectory reflects a kind of compensation that attempts to negate the forces induced by the applied force field and thus inheres to the properties of the force field itself. As velocity along the trajectory increases, the compensation force increases and the after-effect trajectory curves away from the baseline trajectory, thus depicting a mirror image of the before-effect trajectory. Both the compensation and the reoptimization models [4,8] obey to this premise (Figs 4 and S1). Nonetheless, the data presented in this study are incompatible with these models. The expected mirror organization was completely absent in 11/22 participants, and present but far smaller than expected in the remaining participants (Fig 7). Yet, we did not average data across the participants, provided single participant analyses (S3 and S4 Figs), and made the raw data available to give a chance to any possible interpretation.

At this stage, it is interesting to consider the implication of adaptation at the action selection level. It would mean that each new adaptation requires the building of a dedicated control policy which inherits from general motor abilities (e.g. when we hold a manipulandum in a force field task, we do not need to relearn motor coordination from scratch), but remains insulated from general and specific skills (e.g. it does not interfere with our ability to walk or to play the piano). The corresponding motor architecture would come with a heavy computational burden to build, maintain, update, share and exploit each learned ability in each specific context. A solution based on the storage of multiple controllers has been proposed [28], but does not

address the associated computational burden. Furthermore, its proposed implementation through the huge computational power of the cerebellum is probably incompatible with the predominant sensory nature of cerebellar processing [29,30]. Besides these computational issues, interlimb transfer of force field adaptation [31,32] and adaptation by mere observation [33] are also inconsistent with adaptation at the action selection level. The study of de Rugy et al (2012) which is often taken to argue against optimal motor control models, for once, would be consistent with our view [34]. They showed that human participants failed to reoptimize their muscle recruitment patterns following (virtual) changes in muscle actions. They interpreted their results by the existence of "habitual" coordination patterns that are unaffected by selective modifications of the peripheral apparatus. A further interpretation could be that there is no available mechanism for adaptation at the action selection level. A last, indirect argument is related to the contribution of the primary motor cortex (M1) to motor adaptation. Since there is strong evidence that M1 participates to low-level (e.g. muscular) aspects of motor control [35–37], a likely hypothesis is that M1 neurons are involved in processes subserving adaptation to dynamic perturbations (e.g. force field). Yet, Perich and Miller (2017) have shown that the directional selectivity of M1 neurons was modified by the application of a force field but remained unchanged during the course of adaptation [38]. Their results suggest that adaptation occurs upstream of M1 and is transmitted to M1 which is responsible for motor execution.

An alternative view is that adaptation to a novel motor environment relies on changes at the goal selection level (redirection model), i.e. aiming toward appropriately chosen successive spatial goals (e.g. via-points) would mimic adaptation and after-effects in a force field (Fig 8). In the proposed scenario, the "memory" of the perturbation is not a continuous mapping between state and force but a discrete set of via-points. This scenario can be reproduced in a simulation by a multiple-step target jump protocol involving several intermediate via-points and the actual target where the via-points have been chosen by hand to obtain an adapted trajectory which is close to the baseline trajectory (Fig 8). Our data are not incompatible with this view. Yet, they cannot be said to support it since the proposed adaptation mechanism remains incompletely specified: there is no structured approach to select or learn to select proper via-points. Interestingly, the redirection model is versatile enough to account for the whole dataset. The absence of a mirror effect in some participants and its presence in others can be explained by a specific configuration of via-points (Fig 8A and 8B). Whether it is possible to find the location of the via-points for a given task remains an open question. It is tempting to assume that variability should be minimal at the via-points as in the experiment of Todorov and Jordan (2002) [23] in which the via-points are indicated to the participants (their Fig 3). Unfortunately, kinematic variability is so large (e.g. Fig 3G) that we cannot expect anything precise from an analysis of variability, suggesting that in the absence of a constrained trajectory, the via-points may be themselves be subject to variability from trial to trial.

The proposed mechanism should not be linked to the kind of explicit, cognitive strategy that can be used to compensate for a visuomotor rotation simply by changing the aiming direction [39]. Here, the proper choice of via-points is conceived as the outcome of a learning process. What drives the learning process is not specified, but could possibly be cast in a cost/benefit framework (e.g. effort vs accuracy). We have no information on the explicit or implicit nature of the mechanism. Yet, in post-experiment interviews, we noted that several participants believed that the after-effect deviations were due to a force field and not to their own behavior, which suggests that they probably had little conscious control over their behavior once adapted to the perturbation.

A possible concern is the seemingly ad hoc nature of the proposed scenario. However this scenario is derived from a consistent theoretical construct which accounts for the production

of fast and slow movements, the distinction between discrete and rhythmic movements, the ubiquity of isochronous behaviors, the existence of scaling laws, power laws and speed-accuracy tradeoffs [24]. Thus motor control would involve a unique, general-purpose, task-independent action selection mechanism (controller) and each task would have its own representation defined as a series of successive intermediate goals updated at a fixed frequency and pursued at a fixed horizon. In this framework, a skilled movement is not defined by the operation of a dedicated, "skilled" controller, but the use of a dedicated, "skilled" task representation. Consider the following example. It is probable that none of the readers of this article have the tennis skills of a top ranked tennis player. Yet, for the most part, they would not be clumsy in activities of daily living, probably have their own motor skills, and should sometimes be able to produce a magnificent backhand worthy of a good tennis player. Accordingly, the difference between a novice and an expert would not be found at a control level, e.g. a difference in mastering coordination, but at the level of task representation, i.e. how successive goals are consistently set to properly elicit and guide actions. Our view of motor adaptation can effortlessly be cast in this framework. Interestingly, the computational burden associated with the storage of multiple controllers is significantly alleviated with the storage of multiple task representations. Task representations are discrete sets which are much more frugal in neural resources than continuous mappings. Furthermore, they can be scaled spatially and shared between effectors, accounting for motor equivalence. Issues related to the stability and flexibility of skills appear much less enigmatic when skills are conceived as task representations rather than controllers.

The main limitation of this study is its strong reliance on computational modeling. Our conclusions are based on the divergence between experimental data and predictions of the compensation/reoptimization models. So it is fundamental to check that the proposed predictions are both robust and realistic. As far as robustness is concerned, there is no difficulty with the compensation model which is well-formulated and easy to simulate. However, this model has little general relevance for motor control as it does not provide solutions to central problems such as trajectory formation and coordination [23]. For this reason, we have not pursued comparisons with this model. The reoptimization model is based on optimal feedback control [23] and has been updated here to account for proper online feedback control [24]. It generates movements with realistic trajectories, velocity profiles, and amplitude and frequency contents (Fig 3). We have shown that its predictions are robust to parameter changes (S6, S7 and S8 Figs).

An unsettled and interesting issue is related to the intensity of the applied force field. The predictions were obtained with a 2-Ns/m force field as compared to the 10-Ns/m field of the experiment. In the model, for a given perturbation intensity, the size of the lateral deviation of the before-effect trajectory is determined by the interplay between the operation of the state estimator and the dynamics of the arm. Two observations can be made. First, changes in parameters of the state estimator (feedback delay, noise ratio) can reduce the impact of the force field. Yet even a fine tuning of these parameters would not lead to a realistic trajectory deviation for a 10-Ns/m field. Second, parameters of the dynamics have also an influence on the response to perturbations. For instance, we have assessed the influence of the torque ratio (S8 Fig). This parameter reflects the relative efficiency of shoulder and elbow muscles, but its value is not easy to set as it depends on the physiological cross-sectional area, the innervation ratio, the moment arm and the modulation of force production by firing rate and recruitment in pools of motoneurons of each muscle. Furthermore, we cannot play freely with this parameter as it has a strong impact on the timing of the movement (S8D Fig). Other parameters of the dynamics cannot be modified as they pertain to intrinsic characteristics of the arm.

We propose two ideas to obtain quantitatively more realistic deviations with respect to the intensity of the perturbation. The first idea is to use a more realistic dynamics for the modeled arm. For simplicity, we considered the control of a planar two-link arm. Yet the participants were free to use all available degrees of freedom from the trunk to the wrist. The corresponding kinematic chain would likely offer a larger inertial resistance to perturbations. The second idea is in fact an extension of the first one and invokes impedance to account for resistance to perturbations, i.e. not only inertia, but also viscosity and stiffness, could contribute to the resistance [40,41]. In the simulations, we used a long feedback delay (0.12 s) to clearly indicate that any kind of instantaneous, short-latency and medium-latency visco-elastic contributions of muscles and tendons remained unmodeled. A model of these contributions is feasible for perturbations about a static posture [42] but remains elusive for perturbations during ongoing movements. Note that the very efficient elastic feedback along the desired trajectory used in the compensation model cannot be included in the reoptimization model or the redirection model due to the absence of a desired trajectory.

## Materials and methods

### Computational modeling

We simulate displacements of a planar two-link arm whose dynamics are given by

$$\ddot{\theta} = \mathbf{M}(\theta)^{-1}(\tau_u + \tau_e - C(\theta, \dot{\theta})) \tag{1}$$

where $\theta = [\theta_{sh}, \theta_{el}]$ are the shoulder and elbow angles, $\mathbf{M}(\theta)$ the inertia matrix, $C(\theta, \dot{\theta})$ the vector of velocity-dependent torques, $\tau_u$ the control torque produced by actuators and $\tau_e$ the torque due to external forces applied on the arm. We define

$$\mathbf{M}(\theta) = \begin{bmatrix} d_1 + 2d_2 \cos \theta_{el} & d_3 + d_2 \cos \theta_{el} \\ d_3 + d_2 \cos \theta_{el} & d_3 \end{bmatrix}$$

and

$$C(\theta, \dot{\theta}) = d_2 \begin{bmatrix} 2\dot{\theta}_{sh}\dot{\theta}_{el} + \dot{\theta}_{el}^2 \\ -\dot{\theta}_{sh}^2 \end{bmatrix} \sin \theta_{el}$$

where $m_{sh}$ and $m_{el}$ are the link masses, $l_{sh}$ and $l_{el}$ the link lengths, $I_{sh}$ and $I_{el}$ the moments of inertia, $s_{sh}$ and $s_{el}$ the distances from the joint center to the center of mass, $d_1 = I_{sh} + I_{el} + m_{el}l_{sh}^2$, $d_2 = m_{el}l_{sh}s_{el}$ and $d_3 = I_{el}$.

Displacements are perturbed by a velocity-dependent force field producing a force proportional to the velocity along the movement direction $\psi$ (direction is measured relative to initial hand position and 0 is rightward) and perpendicular to this direction. The force field is described by

$$\mathbf{D} = \mathbf{R}(\psi)^{-1} \times \begin{bmatrix} 0 & 0 \\ \varphi & 0 \end{bmatrix} \times \mathbf{R}(\psi) \tag{2}$$

where $\varphi$ is the force level ($\varphi > 0$ for a counterclockwise perturbation, $\varphi < 0$ for a clockwise perturbation) and $\mathbf{R}(\psi)$ the rotation matrix of angle $\psi$. The perturbation torque is

$$\tau_e = \tau_e^{\varphi} = \mathbf{J}(\theta)^{\mathrm{T}} \mathbf{D} \mathbf{J}(\theta)\dot{\theta}$$

where $\mathbf{J}(\theta)$ is the Jacobian matrix of the kinematics

$$\mathbf{J}(\theta) = \begin{bmatrix} -l_{\text{sh}} \sin\theta_{\text{sh}} - l_{\text{el}} \sin(\theta_{\text{sh}} + \theta_{\text{el}}) & -l_{\text{el}} \sin(\theta_{\text{sh}} + \theta_{\text{el}}) \\ l_{\text{sh}} \cos\theta_{\text{sh}} + l_{\text{el}} \cos(\theta_{\text{sh}} + \theta_{\text{el}}) & l_{\text{el}} \cos(\theta_{\text{sh}} + \theta_{\text{el}}) \end{bmatrix}$$

Parameters are: $m_{\text{sh}} = 1.4$ kg, $m_{\text{el}} = 1.1$ kg, $l_{\text{sh}} = 0.3$ m, $l_{\text{el}} = 0.33$ m, $s_{\text{sh}} = 0.11$ m, $s_{\text{el}} = 0.16$ m, $I_{\text{sh}} = 0.025$ kg m$^2$, $I_{\text{el}} = 0.045$ kg m$^2$.

In all the simulations, the initial arm configuration is [45˚, 90˚], movement amplitude is 0.1 m, movement direction is $\psi = 90$˚, and force (field) level is $\varphi = 2$ Ns/m. Four conditions are considered: baseline, in the absence of the force field; before-effect, in the presence of the force field before adaptation; adapted, in the presence of the force field after adaptation; after-effect, in the absence of the force field after adaptation.

## Compensation model

The compensation model is taken from [4]. The principle is the following. First we derive a desired 1-s spatial trajectory for a 0.1-m forward displacement based on a 0.5-s 0.1-m long minimum-jerk trajectory [21] followed by a 0.5-s stationary posture. Second we use the arm inverse kinematics to obtain the desired angular trajectory $\theta^*(t)$, and the arm inverse dynamics (Eq 1) to calculate the joint torques $\tau_u^*(t)$ which produce the desired angular trajectory. Third we obtain actual angular trajectories using

$$\ddot{\theta} = \mathbf{M}(\theta)^{-1}(\tau_u^* + \tau_e + \tau_c - \mathbf{B}(\theta - \theta^*) - C(\theta, \dot{\theta}))$$

where $\tau_c$ is a compensation torque built by adaptation, and $\mathbf{B}$ a feedback gain along the desired trajectory ($\mathbf{B} = 20\mathbb{I}_2$ Nm/rad, where $\mathbb{I}_2$ is the 2×2 identity matrix). The four conditions are: baseline, $\tau_e = \tau_e^0 = \tau_c = 0$; before-effect, $\tau_e = \tau_e^\varphi$ and $\tau_c = 0$; adapted, $\tau_e = \tau_e^\varphi$ and $\tau_c = \tau_e^{-\varphi} = -\tau_e^\varphi$; after-effect, $\tau_e = \tau_e^0 = 0$ and $\tau_c = \tau_e^{-\varphi} = -\tau_e^\varphi$.

## Reoptimization model

The reoptimization model is an extension of the model described in [8]. The control torque $\tau_u = [\tau_u^{\text{sh}}, \tau_u^{\text{el}}]$ is derived from a control input $u = [u_{\text{sh}}, u_{\text{el}}]$ according to

$$\begin{cases} v\dot{\alpha}_i = -\alpha_i + \varepsilon_i \\ v\dot{\varepsilon}_i = -\varepsilon_i + u_i \\ \tau_u^i = g_i\alpha_i \end{cases} \tag{3}$$

where $i = \{\text{sh, el}\}$, $\alpha_i$ is muscle activation, $\varepsilon_i$ muscle excitation, $g_i$ muscle gain and $v$ the muscle time constant (linear second-order muscle model; [43]). We define a state vector $X = [\theta_{\text{sh}}, \theta_{\text{el}}, \dot{\theta}_{\text{sh}}, \dot{\theta}_{\text{el}}, \alpha_{\text{sh}}, \alpha_{\text{el}}, \varepsilon_{\text{sh}}, \varepsilon_{\text{el}}]$ and rewrite the dynamics (Eqs 1 and 3) as $\dot{X} = F_0(X, u) + n_{\text{dyn}}$ for the unperturbed dynamics or $\dot{X} = F_\varphi(X, u) + n_{\text{dyn}}$ for the perturbed dynamics, where $n_{\text{dyn}}$ is additive noise on the dynamics. We formulate an optimal feedback control problem for this dynamics as a search for a control policy $u(t)$ to reach a goal $X = [\theta_{\text{sh}}, \theta_{\text{el}}, \dot{\theta}_{\text{sh}}, \dot{\theta}_{\text{el}}, \alpha_{\text{sh}}, \alpha_{\text{el}}, \varepsilon_{\text{sh}}, \varepsilon_{\text{el}}]$ while minimizing the cost

$$\mathfrak{I}_{F.} = \sum_{i=\text{sh,el}} \int\limits_t^{t+T_{\text{H}}} u_i^2 dt \tag{4}$$

where $F_{\cdot}$ is either $F_0$ or $F_\varphi$ to indicate whether optimization applies to the unperturbed or the

perturbed dynamics, and $T_{\mathrm{H}}$ is the planning horizon [24]. In [8], optimization runs on a fixed duration (0.5 s) and thus cannot be used to simulate before-effect and after-effect conditions which require flexible time to produce online movement corrections. Control with a planning horizon offers an efficient solution to time flexibility as at any time and in any changing situation due to a perturbation there always remains the duration of a planning horizon to reach designated goals [24]. The initial boundary condition is given by $X(t) = \hat{X}(t)$, where $\hat{X}(t)$ is the estimated value of $X(t)$ provided by an optimal state estimator using forward modeling and delayed sensory feedback with delay $\Delta$ [24,44]. The state estimator is given by

$$\dot{\hat{X}}(t) = \hat{F}(\hat{X}(t), u(t)) + \mathbf{K}(t)(y(t - \Delta) - \mathbf{H}\hat{X}(t))$$

where $\hat{F}$ is the dynamics for estimation which is either $F_0$ or $F_\varphi$ (see below),

$$\mathbf{H} = [\mathbb{I}_4 \; \mathbb{O}_4]$$

is a 4×8 observation matrix ($\mathbb{I}_4$ is the 4×4 identity and $\mathbb{O}_4$ the 4×4 null matrix), indicating that only the position and velocity are observed, $\mathbf{K}(t)$ the Kalman gain and

$$y(t) = \mathbf{H}X(t) + n_{\mathrm{obs}}$$

where $n_{\mathrm{obs}}$ is additive observation noise. The Kalman gain is given by

$$\mathbf{K}(t) = \mathbf{A}(t)\mathbf{P}(t)\mathbf{H}^T(\mathbf{H}\mathbf{P}(t)\mathbf{H}^T + \mathbf{\Omega}^\omega)^{-1}$$

where

$$\mathbf{A}(t) = \frac{\partial F_\bullet(t)}{\partial X}$$

and

$$\mathbf{P}(t + \delta) = \mathbf{\Omega}^\xi + (\mathbf{A}(t) - \mathbf{K}(t)\mathbf{H})\mathbf{P}(t)\mathbf{A}(t)^T$$

where $\delta$ is the integration timestep, $\Omega^\omega$ the covariance matrix of observation (sensory) noise $n_{\mathrm{obs}}$ (4-dimensional, zero-mean, Gaussian random vector) and $\Omega^\xi$ the covariance matrix of dynamic (motor) noise $n_{\mathrm{dyn}}$ (8-dimensional, zero-mean, Gaussian random vector). We take

$$\mathbf{\Omega}^\omega = \sigma_\omega \times \mathrm{diag}[1, 1, 10, 10]$$

and

$$\mathbf{\Omega}^\xi = \sigma_\xi \times \mathrm{diag}[1, 1, 10, 10, 100, 100, 1000, 1000]$$

where $\sigma_\omega$ and $\sigma_\xi$ are the variance of sensory and motor noise, respectively, and diag[] indicates the diagonal matrix with listed values on the diagonal. The state estimator is formulated to be optimal taking into account the feedback delay as explained in the Supplementary Notes of [23].

To control movement duration, the goal $X^\#$ is updated every $T_{\mathrm{G}}$ within a series of successive intermediate goals (via-points) $S = \{X_0, X_1, \cdots, X_n\}$ with $X_n = X^*$, i.e. $X^\# = X_0$ at $t = 0$, $X^\# = X_1$ at time $t = T_{\mathrm{G}}, \cdots, X^\# = X_n$ at time $t = nT_{\mathrm{G}}$, where $X^*$ is the final goal of the movement [20,24].

The four conditions are: baseline: $\tau_e = \tau_e^0 = 0$, $\mathfrak{I}_{F_0}$, $\hat{F} = F_0$; before-effect: $\tau_e = \tau_e^\varphi$, $\mathfrak{I}_{F_0}$ (the trajectory is planned based on the unperturbed dynamics but executed against a perturbation), $\hat{F} = F_0$ (the estimator is unaware of the perturbation); adapted, $\tau_e = \tau_e^\varphi$ and $\mathfrak{I}_{F_\varphi}$ (the trajectory is planned based on the perturbed dynamics and executed against a perturbation), $\hat{F} = F_\varphi$ (the

estimator is tuned to the perturbed dynamics); after-effect, $\tau_e = \tau_e^0 = 0$, $\mathfrak{I}_{F_\varphi}$ (the trajectory is planned based on the perturbed dynamics but executed in the absence of the perturbation), $\hat{F} = F_\varphi$ (the estimator remains tuned to the perturbed dynamics). The same series of via-points $S$ is used in all the conditions. The fact that the estimator becomes adapted to the perturbed dynamics is consistent with experimental observations [45,46].

Parameters are: $\nu = 0.05$ s, $g_{sh} = 2$, $g_{el} = 1$, $T_H = 0.28$ s, $T_G = 0.13$ s, $\Delta = 0.12$ s, $\delta = 0.01$ s, $\sigma_\omega = 1$, $\sigma_\xi = 1$. The final goal state is $X^* = [60.7°, 60°, 0,0,0,0,0,0]$, i.e. the final shoulder and elbow angles corresponding to a 0.1-m forward displacement, zero final velocity, activation and excitation.

### Redirection model

The redirection model is taken from [24] and customized to the current formulation. The baseline and before-effect conditions are the same as for the reoptimization model. In the adapted and after-effect conditions, the cost function is $\mathfrak{I}_{F_0}$, i.e. the controller is unaware of the perturbation, but the series of via-points $S$ used in the baseline and before-effect conditions is replaced by a new series of via-points $S'$ which defines adaptation. Like the controller, the estimator remains unaware of the perturbation ($\hat{F} = F_0$).

### Numerical solution

The reoptimization and redirection models are simulated numerically using the iLQR method proposed by Li and Todorov (2004) [47]. For this, we reformulate the optimal control problem defined by Eq 4 and the final boundary constraint $X^{\#}$ as a "regulator" problem with a cost function including both the control cost and the final boundary constraint as a task cost. Parameters $w_u$ (for the control), $w_\theta$, $w_{\dot\theta}$, $w_\alpha$, $w_\varepsilon$ (for via-points) and $w_\theta^*$, $w_{\dot\theta}^*$, $w_\alpha^*$, $w_\varepsilon^*$ (for the final goal) are necessary to weight the different terms of the cost function. The parameters are: $w_u = 0.00001$, $w_\theta^* = 10$, $w_{\dot\theta}^* = 0.1$, $w_\alpha^* = 0.01$, $w_\varepsilon^* = 0.01$, $w_\theta = w_\theta^*$, $w_{\dot\theta} = 0$, $w_\alpha = 0$, $w_\varepsilon = 0$. This means that only the position is constrained at the intermediate goals.

Note that the models are formulated in a stochastic setting (noise on the dynamics and the observation) but are simulated without noise. There is no particular reason to add noise in the simulations.

### Experiment

**Ethics statement.** The experiment was approved by Comité d'Ethique de La Recherche at Sorbonne Université (CER-2021-112). Participants signed a consent form prior to participating in the experiment and in accordance with the ethical guidelines of Sorbonne Université and in accordance with the Declaration of Helsinki.

**Participants.** Twenty-two volunteers (20–30 yr old, 8 female) participated in the behavioral experiment. According to the Edinburgh Protocol of handedness [48], 18 were right-handed, 2 left-handed and 2 ambidextrous. They had no known neurological disorders and normal or corrected to normal vision and they were uninformed as to the purpose of the experiment.

**Apparatus.** Participants were seated on a chair and used their dominant hand (their most comfortable hand for ambidextrous participants) to move the handle of a robotic arm programmed to constrain the displacement of the hand in a horizontal plane and apply force perturbations. Task instructions, feedback information, and continuous visual feedback of hand displacement were provided on a monitor placed vertically in front of the participant. The flow of the task was controlled by a personal computer running Windows 7 (Microsoft Corporation, USA). The 3D position of the robot was recorded at 1000 Hz and stored on the

computer for offline processing and analysis using custom written Matlab scripts (Mathworks, Natick, MA, USA).

**Experimental procedure.** The participants were asked to make forward reaching movements from a start position to a target position located 0.1-m away using visual information displayed on the monitor (start position: 0.6-cm diameter white circle; target position: 1-cm diameter white circle; moving cursor: 0.3-cm diameter black circle). To start a trial, the participants placed the cursor at the start position and began to move when ready. Once the cursor stopped inside the target circle (cartesian velocity < 0.01 m/s), feedback was given regarding desired movement velocity. The circle appeared blue if the movement was deemed too slow (peak velocity along target direction < 0.25 m/s) or red if deemed too fast (peak velocity > 0.35 m/s). No specific constraint was applied to movement accuracy other than the displacement of the cursor to the target circle. The return movement was unconstrained except for the need to stop inside the start circle (cartesian velocity < 0.01 m/s) to start the next trial.

On some trials, a velocity-dependent force field was applied during the forward displacement as defined by Eq 2 with $\psi = 90°$ and $\varphi = \pm10$ Ns/m. The force field was CCW ($\varphi > 0$) for half of the participants. The participants performed four blocks of trials: block 1 (20 trials, 100% vs 0% of null field vs force field), block 2 (200 trials, 90% vs 10%), block 3 (100 trials, 5% vs 95%), block 4 (min 150 trials, max 400 trials, 10% vs 90%). The last block involved many trials to maximize the number of recordings of after-effect trajectories. Yet the participants were offered the possibility to stop the experiment after 150 trials if they felt exhausted or bored. A pause was proposed between each block. Participants were given the following instructions: "Perform forward reaching movements to the target according to the required speed, as indicated by the color code (blue, green, red). You may return to the starting position at your own pace. Make a brief pause in the target and at the starting position and avoid rhythmic back and forth movements. Sometimes the robot may perturb your movement. Whenever it happens, continue to obey to the task instructions". At the start of recording, the participants were already familiar with the robot as they performed unrelated preliminary trials of force and position measurements. The robot was transparent and easy to manipulate.

**Data processing and analysis.** Raw data were used to obtain the planar trajectory of the hand for each trial. A symmetry relative to the start position/target position axis was applied to the trajectories of participants receiving a CCW perturbation. Velocity, acceleration and jerk were calculated numerically from the two-sample difference of the position, velocity and acceleration signals, respectively. Position, velocity, acceleration and jerk were filtered with a fourth-order Butterworth low-pass filter with a cutoff at 10 Hz. Valid trials were detected by a peak velocity along target direction between 0.25 and 0.35 m/s in the forward part of the movement. For each valid trial, the forward trajectory was extracted by detection of movement onset and offset with a velocity threshold of 0.01 m/s and two time-varying quantities were calculated: (1) the angle (counted positive in the CCW direction) of the tangent to the trajectory relative to the line between the start position and the target position; (2) the time derivative of this angle which is closely related to the curvature of the trajectory.

The valid trials were divided into four categories: baseline (trials of block 1), before-effect (perturbed trials of block 2), adapted (perturbed trials of block 4), and after-effect (unperturbed trials of block 4). For each category, mean trajectory, mean angle and mean angle derivative were calculated over the trials.

The rationale for the choice of the filter cutoff frequency is the following. A power spectrum analysis was performed on the unfiltered timeseries using a specific method for short-duration timeseries [49]. The results are shown in S10 Fig for velocity, acceleration and jerk pooled across trials and participants, separately for each category (baseline, before-effect, adapted, after-effect). Much of the power was below 10 Hz.

## Statistical analysis

A classical Student's *t*-test was used to assess the sign of the trajectory angle derivative ($H_0$: = 0 vs $H_1$: $\neq$ 0). A *p*-value < 0.05 was taken to support $H_1$. A *p*-value > 0.05 indicated that we could not reject $H_0$. To assess the status of $H_0$ vs $H_1$ in the latter case, we calculated the Bayes factor $bf_{10}$ which is the ratio between the likelihood of the data under $H_1$ and $H_0$ [50]. Bayes factors were interpreted according to the following table: $1 < bf_{10} < 3$: anecdotal; $bf_{10} > 3$: substantial. The Bayes factors were calculated with the Matlab toolbox FieldTrip (https://www.fieldtriptoolbox.org/; [51]).

## Supporting information

**S1 Fig. Predictions of the compensation model. A**. Simulated trajectories. **B**. Simulated velocity profiles.
(PDF)

**S2 Fig. Time at which the angle derivative of the before-effect trajectory became positive. A**. All participants with mean value (*thick line*) and 25–75 percentiles (*box*). **B**. Data of participant P7 and mean of all the participants. The *black dashed line* is the model prediction.
(PDF)

**S3 Fig. Participants whose behavior is incompatible with the reoptimization model.** Same format as in Fig 5. For $bf_{10}$, the *dotted lines* correspond, from bottom to top, to substantial =, anecdotal =, anecdotal$\neq$, and substantial$\neq$.
(PDF)

**S4 Fig. Participants whose behavior is partially compatible with the reoptimization model.** Same format as S3 Fig.
(PDF)

**S5 Fig. Two participants that failed to improve their behavior with training.** Same format as Fig 5.
(PDF)

**S6 Fig. Parametric study of the model: influence of feedback delay. A**. Before-effect (*red*) and after-effect (*blue*) trajectories. Feedback delay: 0, 0.05, 0.12, 0.15 s; *light* to *dark color*. **B**. Velocity profile. **C**. Trajectory angle. **D**. Angle derivative.
(PDF)

**S7 Fig. Parametric study of the model: influence of noise ratio.** Same format as S6 Fig. Noise ratio $\sigma^{\xi}/\sigma^{\omega}$ (motor/sensory): 0.1, 1, 10, 100; *light* to *dark color*.
(PDF)

**S8 Fig. Parametric study of the model: influence of torque ratio.** Same format as S6 Fig. Muscle gain ratio $g_{sh}/g_{el}$ (shoulder/elbow): 1, 2, 5, 10; *light* to *dark color*.
(PDF)

**S9 Fig. Parametric study of the model: influence of boundary conditions. A**. Mean and 25–75 percentiles of positive acceleration peaks for baseline (*black*) and before-effect (*red*) trajectories for different boundary conditions at via-points: *p*: only position; *pv*: position and velocity; *pva*: position, velocity and activation; *pvae*: position, velocity, activation and excitation. **B**. Same as **A** for jerk.
(PDF)

**S10 Fig. Power spectrum analysis. A**. Power spectrum density (arbitrary unit) of velocity average across trials and participants, for baseline (*black*), before-effect (*red*), adapted (*green*) and after-effect (*green*) trials. **B**. Same as **A** for acceleration. **C**. Same as **A** for jerk. (PDF)

## Author Contributions

**Conceptualization:** Etienne Moullet, Agnès Roby-Brami, Emmanuel Guigon.

**Data curation:** Etienne Moullet, Emmanuel Guigon.

**Formal analysis:** Etienne Moullet, Agnès Roby-Brami, Emmanuel Guigon.

**Investigation:** Emmanuel Guigon.

**Methodology:** Etienne Moullet.

**Project administration:** Agnès Roby-Brami, Emmanuel Guigon.

**Supervision:** Emmanuel Guigon.

**Validation:** Emmanuel Guigon.

**Writing – original draft:** Etienne Moullet, Agnès Roby-Brami, Emmanuel Guigon.

**Writing – review & editing:** Etienne Moullet, Agnès Roby-Brami, Emmanuel Guigon.

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
