## [Decision Letter · Decision Letter 0]

20 Jun 2022

Dear Dr. Guigon,

Thank you very much for submitting your manuscript "What is the nature of motor adaptation to dynamic perturbations?" for consideration at PLOS Computational Biology.

As with all papers reviewed by the journal, your manuscript was reviewed by members of the editorial board and by several independent reviewers. In light of the reviews (below this email), we would like to invite the resubmission of a significantly-revised version that takes into account the reviewers' comments.

The reviewers were divided in their assessment of the manuscript; I opted to give you the benefit of the doubt. In order to fully address the issues raised, you will need to defend the adequacy of the data to test your hypothesis, and to more comprehensively evaluate alternative hypotheses.

We cannot make any decision about publication until we have seen the revised manuscript and your response to the reviewers' comments. Your revised manuscript is also likely to be sent to reviewers for further evaluation.

Sincerely,

Samuel J. Gershman

Deputy Editor

PLOS Computational Biology

Reviewer's Responses to Questions

**Comments to the Authors:**

Reviewer #1: The authors describe a study aimed at understanding the nature of force field adaptation in reaching movements in humans. They combined empirical studies of FF reaching adaptation with an unusual data analysis technique to distinguish between a model of adaptation that involves learning novel patterns of forces, from a model that instead involves remapping the location of the endpoint target.

I find the central premise/rationale of the study to be highly problematic. The authors try to draw a distinction between FF learning involving learning a new mapping between states and compensatory forces, and a second view that involves modifying the mapping between intended and actual goals 'irrespective of how to achieve these goals' and they use visuomotor rotation as an example of the latter. (Introduction, second paragraph). It's important to note that in visuomotor adaptation, if participants learn to aim to a different target location, that new actual movement direction does in fact involve new patterns of forces as well. Thus I don't see how one can in principle distinguish between these two models, at least not without some empirical method of disentangling the movement goal from the underlying forces.

The authors state (Introduction) that they have not found any study that quantitatively documents the shape of the after-effect trajectories. I think Bhushan & Shadmehr (1999) have done this, no?

- Bhushan N (1998) A Computational Approach to Human Adaptive Motor Control Shadmehr R, ed. Available at: http://citeseerx.ist.psu.edu/viewdoc/download?doi=10.1.1.392.2697&rep=rep1&type=pdf.

- Bhushan N, Shadmehr R (1999) Computational nature of human adaptive control during learning of reaching movements in force fields. Biol Cybern 81:39–60 Available at: http://dx.doi.org/10.1007/s004220050543.

The authors state (Introduction) that after-effect trajectories seem to resemble "kinematic" trajectories rather than "dynamic" trajectories. I have no idea what this means, which is problematic since this is part of the stated premise/rationale of the study.

The authors use a rather unusual method of frequency analysis of kinematics (velocity, acceleration and jerk) in order to distinguish between the competing models described above. I think this is highly problematic, namely using kinematics alone to try to distinguish between models that learn to reach to targets in a force field. It would seem to me the only way to test different models of FF adaptation would be to use models that actually produce forces, e.g. a dynamic model of a two-link planar arm, with for example a hill-type muscle model, and put that model in a simulated FF, and test the two models of control described in the Introduction.

In fact, using an explicit computational model as described above would enable the reader to fully understand what exactly these competing models are, because in such a computational model one would have to make them perfectly explicit in order to actually run the models. This would be my major suggestion for the authors—use a mechanistic model of arm movement that actually generates forces to test competing models of control/learning.

Reviewer #2: I would like to disclose my identity as Jonathan Tsay.

Moullet and colleagues asked, “what is the nature of motor adaptation to dynamic perturbations”. They took a fine-grain approach to analyze after-effect trajectories following force-field perturbations and noticed that the data are not consistent with the predictions of a compensatory mechanism in which a mapping was learned between the goal and the optimal forces to achieve the goals in the presence of the applied forces (e.g., after-effects are not a mirror reflection of before-effects). Instead, the after-effects are consistent with a redirection process where participants (implicitly) aim towards a series of spatially remapped targets (i.e., via-points). The fine-grain analysis of after-effect trajectories is rigorous, but the current manuscript seems to miss several important components (see below):

First, the current version of the manuscript is difficult to follow, and the theoretical differences/underpinnings of the models can be clearer. E.g., which level of behavior do these models try to explain with regards to feedback and feedforward corrections? Some work has suggested that feedforward corrections during late adaptation/aftereffects is a time-shifted version of the feedback corrections during before-effects (Albert & Shadmehr, 2016), whereas others have challenged this idea, positing that feedforward and feedback appear to be more independent (https://drive.google.com/file/d/1Nlhl59X1U8ELzKLTicW1VQSNSCYIRcHd/view). The current manuscript does not address this critical issue.

The authors bring up that the models differ in how they alter different stages of movement (e.g., action selection, goal selection) – however, this terminology can be used differently across papers. For example, some refer action selection as where people explicitly aim (Kim et al., 2020; Krakauer et al., 2019), but in this paper, goal selection seems to correspond to where people explicitly aim. Clarifying how these terms are used in this paper would help readers follow the logic/differences among these models (a schematic may help).

Second, missing model comparisons. The authors have focused on whether different models can qualitatively predict the after-effect trajectories. However, have the authors addressed the differences in model complexity (e.g., number of parameters, or number of assumptions of these models)? The “winning” model (redirection model) includes the notion of via-points, each of which make up the entire trajectory. Do the number of via-points change the model complexity, and therefore, make the re-direction model less parsimonious? An analysis of AIC and BIC may be useful to arbitrate among these models.

Third, missing tests of alternative hypotheses. The current conclusions are largely based on one observation, the trajectory of the aftereffect data. While the re-direction model does a decent job capturing this behavior (but perhaps not parsimoniously, see Point 2), the data may also be consistent with one or more of the following:

a) A model of Proprioceptive re-alignment (or PReMo; see Feature 6 of (Tsay et al., 2021)): participants following forcefield perturbation may experience a shift in their sensed hand position towards the direction of the force-field (Ostry et al., 2010). PReMo provides a theoretical account for the feedforward corrections (although feedback corrections are not considered and thought to arise from another learning system).

b) The presence of cognitive re-aiming strategies: The authors note in the Discussion that the re-direction model does not arise from a change in explicit re-aiming in response to a force-field. Can the authors rule out this explanation? There have been a study that shows explicit re-aiming in response to forcefields (Schween et al., 2020). Aftereffects may also arise from explicit re-aiming if instructions are ambiguous (e.g., “reach directly to the target” could either refer to brining the hand or the invisible cursor to the target; if referring to the invisible cursor, participants may still explicitly re-aim away from the original target during the washout aftereffect period).

c) Before-effects time-shifted to form after-effects: Can the compensatory model be salvaged if a gain (or time shift) parameter was included? That is, can the authors experimentally or logically rule out the notion that after-effects are just a time-shifted version of the before-effects (i.e., people are selecting the same goal, that is, the original target), rather than a re-direction towards another target?

d) while the re-direction is a viable hypothesis, the critical idea (i.e., people are redirecting their movements to spatially re-mapped targets) was not directly tested (currently, only inferred via model comparison). Is there a way to probe where people are aiming, or the location of these via-points directly? If it’s explicit, then the authors could consider using an aiming wheel + aim report method. If these via points are implicit, does the re-direction model generate any qualitatively different predictions with regards to how aftereffects generalize to local/global targets, compared to the compensation hypothesis?

References:

Albert, S. T., & Shadmehr, R. (2016). The neural feedback response to error as a teaching signal for the motor learning system. The Journal of Neuroscience: The Official Journal of the Society for Neuroscience, 36(17), 4832–4845.

Kim, H. E., Avraham, G., & Ivry, R. B. (2020). The Psychology of Reaching: Action Selection, Movement Implementation, and Sensorimotor Learning. Annual Review of Psychology. https://doi.org/10.1146/annurev-psych-010419-051053

Krakauer, J., Hadjiosif, A. M., Xu, J., Wong, A. L., & Haith, A. M. (2019). Motor Learning. Comprehensive Physiology, 9(2), 613–663.

Ostry, D. J., Darainy, M., Mattar, A. A. G., Wong, J., & Gribble, P. L. (2010). Somatosensory plasticity and motor learning. The Journal of Neuroscience: The Official Journal of the Society for Neuroscience, 30(15), 5384–5393.

Schween, R., McDougle, S. D., Hegele, M., & Taylor, J. A. (2020). Assessing explicit strategies in force field adaptation. Journal of Neurophysiology, 123(4), 1552–1565.

Tsay, J. S., Kim, H. E., Haith, A. M., & Ivry, R. B. (2021). Proprioceptive Re-alignment drives Implicit Sensorimotor Adaptation. In bioRxiv. https://doi.org/10.1101/2021.12.21.473747

Reviewer #3: This study examined whether motor adaptation to a dynamic (force) perturbation reflects changes at the action selection level vs at the goal selection level. The authors performed a strict examination of arm trajectories as subjects made reaching movements with one arm to a visuomotor target. A velocity-dependent force field disrupted the reaching movements. The "change in action selection" model predicts mirror-symmetric shapes for the arm trajectories early in adaptation and the after-effect trajectories. This was not found, however. Instead, the afer-effect trajectories suggest that the process of adaptation occurs instead at the goal selection level. By selecting a new goal, the motor system has the option to generate arbitrary trajectories, which in some cases might be more efficient.

I have no problems with the manuscript. The topic is important as it sheds light on how much the motor system controls trajectory details--more than was expected. The approach is sound, the results are clear, and the figures are great. The results are interesting and they will help advance our understanding of human gait control.

I have one comment regarding a premise of the study's hypothesis:

p. 4, par. 2. "after-effect trajectories, ... should incorporate a negative image of the forces induced by the applied force field..." A 2015 study by Krakauer's group showed that the amount of after-effect in rotation learning can be influenced, and even entirely cancelled, by the number of trials performed at asymptote. The interpretation was that reward-related components of motor adaptation.

Minor suggestions:

Minor changes are needed concerning several instances of awkward word choice or use of prepositions. There are many phrases, throughout the manuscript, that are either grammatically incorrect or that vary from typical English usage. I give a few examples below, but this list is not complete. It would be best to have the manuscript revised by a reader with fluent mastery of English.

Examples:

p. 2, par. 2: delete "against reality."

p. 2, par. 2: Change "The results suggest to change our mind..." to something like "The results change our view of motor adaptation."

p. 3, par. 1: change "cooperation" with something like "balance"; change "mandatory" to "necessary."; change "outrageously" to "disproportionately."; delete "upon request."

**Have the authors made all data and (if applicable) computational code underlying the findings in their manuscript fully available?**

Reviewer #1: None

Reviewer #2: Yes

Reviewer #3: **No: **

PLOS authors have the option to publish the peer review history of their article (what does this mean?). If published, this will include your full peer review and any attached files.

Reviewer #1: No

Reviewer #2: **Yes: **Jonathan Tsay

Reviewer #3: No
---

## [Decision Letter · Decision Letter 1]

4 Aug 2022

Dear Dr. Guigon,

We are pleased to inform you that your manuscript 'What is the nature of motor adaptation to dynamic perturbations?' has been provisionally accepted for publication in PLOS Computational Biology.

Best regards,

Samuel J. Gershman

Deputy Editor

PLOS Computational Biology

Reviewer's Responses to Questions

**Comments to the Authors:**

Reviewer #2: The authors have sufficiently addressed my comments. That being said, while I appreciate that the central premise of the paper was to reject the re-optimization model, their results - as the authors noted themselves -- rely heavily on modeling without further experimental validation. I encourage the authors to at a minimum discuss concretely how they/the field should test the redirection model experimentally.

**Have the authors made all data and (if applicable) computational code underlying the findings in their manuscript fully available?**

Reviewer #2: Yes

PLOS authors have the option to publish the peer review history of their article (what does this mean?). If published, this will include your full peer review and any attached files.

Reviewer #2: **Yes: **Jonathan S. Tsay

---

## [Editor Report · Acceptance letter]

24 Aug 2022

PCOMPBIOL-D-22-00735R1 

What is the nature of motor adaptation to dynamic perturbations?

Dear Dr Guigon,

I am pleased to inform you that your manuscript has been formally accepted for publication in PLOS Computational Biology. Your manuscript is now with our production department and you will be notified of the publication date in due course.

With kind regards,

Zsofia Freund
